# Appropriate complementary feeding practices and associated factors among mothers with infants aged 6–8 months in West Gojjam Zone, Northwest Ethiopia: A mixed methods study

Shiferaw Birhanu[1]*, Getu Degu Alene[2], Yeshalem Mulugeta Demilew[2]

**1** School of Health Sciences, College of Medicine and Health Sciences, Bahir Dar University, Bahir Dar, Ethiopia, **2** School of Public Health, College of Medicine and Health Sciences, Bahir Dar University, Bahir Dar, Ethiopia

* yonishife@yahoo.com

## Abstract

### Background

Appropriate complementary feeding is critical to improve children's nutrition, health, and development. However, these practices are notably low among Ethiopian mothers. Therefore, this study aimed to assess appropriate complementary feeding practices and associated factors among mothers with infants aged 6–8 months in West Gojjam Zone, Northwest Ethiopia.

### Methods

A community-based cross-sectional study including 802 mothers with infants aged 6–8 months was conducted from June to December 2023. Study participants were selected using a cluster sampling technique. Quantitative data were collected using structured questionnaires, while semi-structured interview guides were employed for qualitative data collection. Multivariable logistic regression was used to identify factors associated with appropriate complementary feeding practices. A p-value <0.05 was considered statistically significant. Qualitative data were thematically analyzed using Open Code 4.03 software.

### Results

The prevalence rate of appropriate complementary feeding practices among mothers with infants aged 6–8 months was only 9.6% (0.096; 95% CI: 0.077, 0.119). A month increase in the infant's age [AOR = 2.92, 95% CI: (1.99, 4.29)], postnatal counseling on complementary feeding [AOR = 2.64, 95% CI: (1.46, 4.75)], feeding animal-source foods on fasting days [AOR = 2.60, 95% CI: (1.20, 5.66)], higher household wealth: being rich [AOR = 3.13, 95% CI: (1.32, 7.40)], and richest [AOR = 3.16, 95% CI: (1.34,

**Data availability statement:** All relevant data are within the paper and its Supporting Information files.

**Funding:** The author(s) received no specific funding for this work.

**Competing interests:** The authors have declared that no competing interests exist.

7.49)], and perceived susceptibility [AOR = 2.45, 95% CI: (1.39, 4.31)] were predictors of appropriate complementary feeding practices. Additionally, excessive workload, misconceptions, and poverty were barriers to these practices.

## Conclusion

Most mothers in this study practiced inappropriate complementary feeding. Therefore, healthcare providers should strengthen postnatal counseling on complementary feeding and promote provision of age-appropriate animal-source foods on fasting days. Improving households' economic status and mothers' understanding of the risks associated with inappropriate complementary feeding practices is crucial. Collaboration among stakeholders, including women's affairs and religious leaders, can help reduce mothers' workload and address misconceptions about complementary feeding practices.

## Introduction

Complementary food refers to nutritious solid or semi-solid foods provided to young children in addition to breast milk to promote optimal growth and development [1]. The World Health Organization (WHO) recommends introducing complementary foods (CFs) at 6–8 months of age and continuing until 23 months or beyond [2], as breast milk only cannot meet all nutritional needs [3]. Gradually increasing the consistency, frequency, and variety of CFs, while maintaining breastfeeding, is essential to bridge these gaps [4,5]. It is recommended to provide 2–3 meals daily for infants aged 6–8 months, and increasing to 3–4 meals plus snacks for children aged 9–23 months [2,4].

Conversely, inappropriate complementary feeding practices including introducing foods before the recommended age, inadequate feeding frequency, and lack of divers diets are associated with adverse nutritional outcomes such as stunting, wasting and underweight [6]. Early childhood stunting, for example, can lead to permanent impairment in growth and cognitive function, and increase the risk of illness and death [7].

In 2019, two-thirds of children aged 6–23 months globally did not fulfill the minimum meal frequency (MMF) and only one in five rural children had a diverse diet [8]. By 2023, fewer than a quarter of children aged 6–23 months in many countries met a minimum acceptable diet (MAD) [4]. In developing countries, many young children do not meet WHO feeding standards [9–11]. Studies in Ethiopia indicate that complementary feeding practices (CFPs) are inadequate to meet children's energy and nutrient needs [12,13]. The prevalence of appropriate complementary feeding practices (ACFPs) varies significantly, ranging from 8.5% in Damot Weydie district to 56.5% in Lasta district [10,14–19].

Most previous studies assessed minimum dietary diversity (MDD) using the 2008 WHO infant and young child feeding (IYCF) indicators, where MDD is considered met if four or more of the seven food groups are included [20]. The updated 2021 WHO indicators now incorporate breast milk as an eighth food group in MDD calculations,

with the threshold for adequacy raised to consumption of five or more food groups. Evidence suggests that incorporating breast milk into the MDD calculation results in a lower prevalence compared to the 2008 WHO IYCF indicators [21,22]. Additionally, the revised WHO guidelines now include egg and/or flesh food consumption as a complementary feeding indicator [2]. This nutrient-dense food group is crucial for maximizing child nutrition [5] and vital for optimal linear growth [2]. Consequently, this study assessed the ACFPs of mothers for their infants using four indicators: timely introduction of solid, semi-solid, or soft foods (SSSF), MDD, MMF, and egg and/or flesh food consumption.

Efforts have been made in alignment with both the Millennium and Sustainable Development Goals. For instance, in 2003, the WHO developed a strategy for optimal IYCF practices to combat childhood undernutrition [23]. In 2008, it endorsed IYCF indicators [20], and in 2021, it revised these indicators to help countries meet the 2025 and 2030 targets [2]. The Ethiopian government adopted WHO IYCF guidelines and developed a national strategy to improve IYCF practices [24]. Building on these efforts, Ethiopia launched the Seqota Declaration of Zero Hunger in 2015, committing to "end stunting in children under two by 2030" [25]. Despite these efforts, improvements in child health have been noted [9,26], but ACFPs of mothers for their children remains very low in the country.

Existing literature identifies multiple predictors of ACFPs, including sociodemographic characteristics, household economic status, maternal education and occupation, health service utilization [9,15,17–19,27–29], child age [10,16–18,29], child sex, access to CFPs information [27], family size [18], birth interval [19,30], and maternal attitudes toward CFPs [16]. Conversely, inappropriate CFPs are associated with low household income, low maternal education [31], early marriage, time-use conflicts, poverty [32] food insecurity, and home deliveries [30].

Child feeding practices are also influenced by psychological and social factors [33,34], which were not adequately addressed in our context. This study contributes to the literature by examining the psychological (predisposing) factors affecting child feeding practices. These factors include perceived susceptibility to and severity of the consequences of inappropriate feeding, perceived benefits, and barriers to providing appropriate CFs. Furthermore, previous studies in Ethiopia assessed factors affecting CFPs using quantitative methods [10,15,16,18,19], but failed to explore the underlying reasons (the "how" and "why"). Therefore, this study incorporated qualitative approaches to explore the barriers to ACFPs.

Although the WHO recommends continuing complementary feeding up to 23 months, this study specifically focused on mothers with infants aged 6–8 months. The narrower age range was chosen because it is when mothers typically begin introducing CFs. During this time, many mothers fear providing diverse diets, believing their children do not need more variety. The recommendation for intervention, given that this practice in this age group contributes to undernutrition, should be supported by evidence. Additionally, these mothers with infants serve as the baseline population for the intervention titled "Effects of complementary feeding counseling on appropriate complementary feeding practices and child undernutrition in West Gojjam Zone, Northwest Ethiopia" (registered at ClinicalTrials.gov, NCT05871346). Therefore, this study aimed to assess appropriate complementary feeding practices and associated factors among mothers with infants aged 6–8 months in West Gojjam Zone, Northwest Ethiopia. The findings will provide evidence for policymakers and program planners to develop targeted counseling interventions, thereby improving IYCF practices. Additionally, the results will support healthcare providers in enhancing their counseling strategies.

## Materials and methods

### Study setting and period

This study was conducted in the rural areas of the West Gojjam Zone, in the Amhara region of Ethiopia, 389 km northwest of Addis Ababa. The zone comprises 22 districts, including 14 rural and 8 urban. As of 2023, the total population was estimated at 2,833,067, with 2,337,280 (82.5%) living in rural areas. Females made up 50.33% of the population, while infants accounted for 3.11%. There were 445 *kebeles* (the smallest administrative units in Ethiopia), of which 395 (88.8%) were rural. Approximately 72,690 infants resided in these rural *kebeles* (calculated as 2,337,280 x 3.11%) [35,36]. The study was conducted from June to December 2023.

## Study design

A community-based concurrent descriptive mixed-methods study was conducted in West Gojjam Zone, Northwest Ethiopia. Qualitative data were collected alongside quantitative data to address the "how" and "why" questions that quantitative methods alone cannot answer. Both the quantitative and qualitative methods were described in detail.

## The quantitative study

### Study population

The source population included all mothers with infants aged 6–8 months in the zone, while the study population comprised mothers with infants aged 6–8 months from randomly selected *kebeles*.

### Inclusion and exclusion criteria

Since this study served as the baseline for a cluster randomized controlled trial (ClinicalTrials.gov, NCT05871346), the inclusion criteria comprised mothers with infants aged 6–8 months who had resided in the study area for at least 6 months prior to the survey. Mothers intending to leave the study area before completion of the study were excluded.

### Sample size determination

To calculate the sample size, we used a 30% prevalence of ACFPs among mothers with children aged 6–23 months, based on a study conducted in Shashemene, Ethiopia [16]. Both the single population proportion formula and Epi Info's sample size calculation method (which incorporates predictor variables) were applied, and the largest value was selected. Using the single population proportion formula ($n = \frac{\left(Z_{\frac{\alpha}{2}}\right)^2 p(1-p)}{\in^2}$), we obtained a sample size of 679, assuming a 95% confidence interval, 5% margin of error, design effect (DE) of 2, and a 5% non-response rate. The sample size calculated using predictor variables such as child's age, socioeconomic status, number of antenatal care (ANC) visits, information sources, and mothers' attitudes and knowledge [16], in Epi Info version 7, assuming a 95% confidence interval, 80% power (1-β), a 1:1 ratio of unexposed to exposed groups, and a 5% non-response rate, yielded a sample size of 781. However, as this study used cluster randomization (NCT05871346), the sample size calculated to assess undernutrition, using the predictor variables of underweight (sex and age of the child) [37], resulted in the largest sample size of 802.

### Sampling procedure

A cluster sampling technique was used to select mothers with infants aged 6–8 months. From the 22 zonal districts, nine were excluded: one rural district (due to an ongoing nutrition intervention program) and eight urban districts (to maintain study homogeneity). From the remaining 13 rural districts, four (Dembecha Zuria, Jabitehenan, Bure Zuria, and Yilmana Densa Woreda) were randomly selected using simple random sampling (SRS). *Kebeles* were considered as clusters from which participants were drawn. We proportionally allocated and randomly selected 30 *kebeles* through SRS: 8 from Dembecha Zuria, 8 from Jabitehenan, 6 from Bure Zuria, and 8 from Yilmana Densa. Eligible mothers with infants in each *kebele* were screened through a house-to-house survey to confirm eligibility and their willingness to participate. Within the selected clusters, all mothers with infants who met the inclusion criteria were recruited to minimize selection bias through complete enumeration.

### Definition of terms

- Timely introduction of complementary foods: refers to when mothers initiated complementary feeding within 6–8 months of their infants' age. It was considered untimely if CFs were introduced before six months [2].

- Minimum dietary diversity (MDD): was achieved when mothers fed infants at least five out of the following eight food groups during the previous day: 1) breast milk; 2) grains, roots, tubers, and plantains; 3) pulses (beans, peas, lentils),

nuts, and seeds; 4) dairy products (milk, infant formula, yoghurt, cheese); 5) flesh foods (meat, fish, poultry, organ meat); 6) eggs; 7) vitamin A-rich fruits and vegetables; and 8) other fruits and vegetables [2]. MDD was not met when mothers fed infants fewer than five food groups [2].

- Minimum meal frequency (MMF): This was achieved when breastfeeding infants aged 6–8 months consumed solid, semi-solid, or soft foods (SSSF) at least two times during the previous day. Infants did not meet MMF when they consumed SSSF fewer than two times during the previous day [2].

- Minimum acceptable diet (MAD): This was met when breastfeeding infants aged 6–8 months satisfied the MDD and MMF during the previous day. It was not met when either MDD or MMF was not satisfied [2].

- Egg and/or flesh food consumption: This was considered met when infants consumed either egg and/or flesh foods (meat, or fish, or poultry, or organ meat) during the previous day. It was not met when infants did not consume egg and/or flesh food during the previous day [2].

- Appropriate complementary feeding practices (ACFPs): This was defined as meeting all four indicators: timely introduction of SSSF, MDD, MMF, and egg and/or flesh food consumption. The practice was considered inappropriate if any one of these indicators was not met [2].

## Data collection instruments

Data were collected using an interviewer-administered questionnaire adapted from the 2019 Ethiopia Mini Demographic Health Survey (EMDHS), the revised WHO IYCF indicators (2021), and other similar studies [10,16–18,38]. The questionnaire included sociodemographic and economic characteristics, child characteristics, CFPs, health service utilization, questions assessing perceptions, knowledge, and attitudes. To ensure participant comprehension, the English questionnaire was translated into Amharic (the local language).

## Data collection methods

Eight trained health extension workers collected the data from mothers in their homes, obtaining written informed consent or fingerprints from those who could not read or write. Four health professionals with master's degrees supervised the data collectors. A 24-hour dietary recall survey, tailored to local food availability, was used to gather information on mothers' feeding practices for their infants. The recall period for data collection starts at the beginning of the survey and extends 24 hours back (Fig 1).

## Dependent and independent variables

**Dependent variable.** Complementary feeding practices (Appropriate, Inappropriate).
**Independent variables.** The independent variables included the following: sociodemographic and economic characteristics (maternal age, religion, education, occupation, marital status, decision-making, family size, child's sex and age, number of children under five, wealth index), health service utilization (history and frequency of ANC visits, place of delivery, history and frequency of postnatal care (PNC) visits, complementary feeding (CF) counseling during PNC visits, birth order and birth interval), recent child morbidity (diarrhea, fever, or difficulty breathing), other maternal factors (knowledge, attitudes and perceptions toward ACFPs).

## Measurements

The household wealth index was determined using principal components analysis (PCA) based on Ethiopia's Demographic Health Survey assessment items, including housing conditions, drinking water sources, latrines, household assets, fuel types, livestock, and land ownership [38]. To transform categorical data into a binary format, the responses

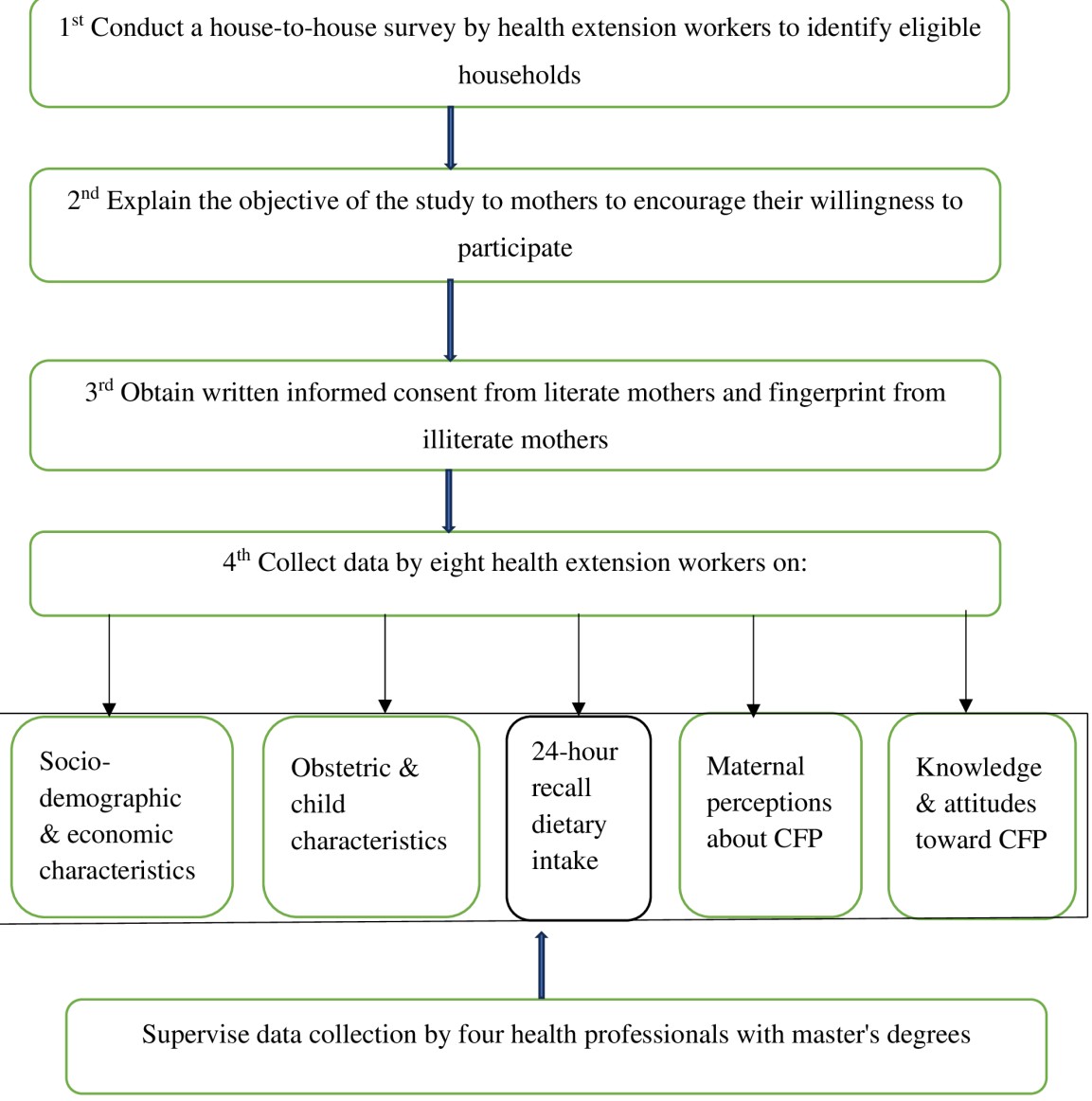

**Fig 1. Data collection methods flow chart. CFPs. Complementary feeding practices.**

of the non-dummy variables were classified into three categories, with the highest score coded as 1 and the two lower scores as 0. Assumptions for PCA were checked, and all variables included in the model were dichotomized, making the data more manageable for analysis and facilitate the interpretation of findings [39]. Following varimax rotations to make the components more interpretable, eleven household asset variables with a commonality value greater than 0.5 were retained to produce factor scores. These include: 1) radio, 2) dining table, 3) sofa or chairs, 4) bed with cotton or sponge mattress, 5) cattle (cows or bulls), 6) horses, donkeys, or mules, 7) chickens, 8) watch (by at least one household member), 9) animal-drawn cart (by at least one household member), 10) location of drinking water, and 11) proximity of drinking water from home. The first principal component accounted for 18.65% of the variance, while the first six components collectively explained a significant portion (71.8%) of the total variance. Wealth quintiles were used to examine

how equitably other indicators were distributed by wealth status. Quintiles of the wealth score were created to categorize households as poorest, poor, medium, rich, and richest [40].

Mothers' knowledge of ACFPs was assessed using 10 positive questions. Correct responses were coded as 1, while incorrect responses coded as 0. Mothers scoring 70% or higher were classified as having good knowledge, those scoring between 51% to 69% as having average, and those scoring 50% or lower as having poor knowledge [41].

Mothers' attitudes toward ACFPs were assessed using 10 Likert-scale questions. Responses were summed to create a total attitude score and categorized into three groups: negative (lowest tertile), uncertain (middle tertile), and positive (highest tertile). For analysis, the highest tertile was classified as a positive attitude (coded as 1), while the remaining two tertiles were combined into a negative attitude (coded as 0) [39,41].

The predisposing factors, including perceived susceptibility to and severity of undernutrition, as well as the benefits, and barriers of CFPs were assessed using a validated tool adapted from a previous study conducted in the rural setting in Dessie, Amhara regional state, Ethiopia, in accordance to child feeding practices. According to the report of that study, the internal consistency (Cronbach's α) for each subscale was greater than 0.7: perceived susceptibility = 0.80, perceived severity = 0.88, perceived benefits = 0.82, and perceived barriers = 0.87 [38]. These factors were assessed by summing their respective composite questions, with responses provided on a 5-point Likert scale (1 = strongly disagree to 5 = strongly agree). The mean for each variable was calculated. Mothers scoring above the mean were considered to have high perceived susceptibility, severity, benefits, and barriers, while those scoring at or below the mean were considered to have low levels.

## Data quality assurance

The questionnaire was adapted from standard data collection instruments. To ensure consistency, the instrument was first written in English, translated into Amharic, and then back-translated into English. Data collectors and supervisors were selected based on their prior experience in data collection and supervision. A two-day training session was provided for both data collectors and supervisors, covering the content, aims, sampling techniques, and ethical issues before data collection. A pretest assessed the tool's consistency, and amendments were made based on the results. Supervisors closely monitored the data collection, ensured data completeness, and addressed any issues that arose. Data quality was maintained through proper recruitment, adequate training, and thorough supervision of data collectors.

## Data processing and analysis

Before further statistical analyses, assumptions were checked using standard procedures. Manual data coding and cleaning were carried out to identify inconsistencies and missing values. The data were entered in EpiData software version 4.6 and then exported to SPSS 26 for analysis. Descriptive statistics such as frequency distribution and percentages were used to describe the study participants. The prevalence of the introduction of SSSF, MDD, MMF, MAD, egg and/or flesh food consumption, and ACFPs was assessed.

In this study, cluster sampling was implemented to account for the hierarchical structure of the data (mothers with infants aged 6–8 months within *kebeles*). Given the categorical nature of the outcome variable, a generalized linear mixed model (GLMM) was initially fitted to account for cluster-level variables. In this analysis, the intercept only model estimated the intercept at 0.349 (p = 0.089), yielding an intra-class correlation coefficient (ICC) of 0.096. This ICC, which is relatively close to zero (0.096), indicates that 90.4% of the variation in the CFPs was explained by individual-level variables (mothers' characteristics). The calculated sample size was multiplied by a design effect of 2 to better account for the potential impact of clustering on estimates and to increase the overall sample [42]. Based on these findings, we determined that conventional bivariable, and multivariable logistic regression analyses were appropriate for identifying factors affecting ACFPs.

Multicollinearity among predictor variables was assessed using variance inflation factor (VIF) and tolerance values. The VIF ranged from 1.007 to 1.052, while tolerance values ranged from 0.951 to 0.993, indicating no significant multicollinearity. Model fitness was evaluated using the Hosmer-Lemeshow goodness-of-fit test (p=0.881). Variables with p-values <0.25 in the bivariable analysis were included in the multivariable model, using forward stepwise selection to assess their independent effects. In the multivariable analysis, variables with a p-value <0.05 at 95% confidence intervals were considered statistically significant.

## The qualitative study

### Study design and setting

A descriptive qualitative study was conducted parallel to the quantitative study to explore the barriers and enablers of ACFPs in West Gojjam Zone, Northwest Ethiopia. The details of the study setting are described in the quantitative section.

### Population and sampling technique

A purposive sampling technique was used to select fifteen respondents, including four in-depth and eleven key informant interviews. The four in-depth interviews were conducted among mothers with children aged 6–23 months. The eleven key informants comprised four Women Development Army leaders (team leaders of the one-to-five network, a lower-level government structure containing six mothers in Ethiopia), four health extension workers (primary health workers at the *kebele* level), and three health extension officers (professionals assigned to woreda health offices who coordinate health extension workers). All participants were married, Orthodox Christians by religion, and of Amhara ethnicity. Their ages ranged from 22 to 57 years, and their education levels varied from no formal education to a second degree.

### Data collection

Data were collected using an open-ended, semi-structured interview guide developed based on the research questions. First, the guide was prepared in English, then translated into Amharic, the local language, and back-translated into English to ensure consistency. The interview guides consisted of four main parts (S1 Appendix). A digital voice recorder was used to collect the data after obtaining verbal consent. All interviews were conducted by the principal investigator in nearby health center meeting rooms at convenient times. The researcher actively listened to the respondents and asked probing questions. In-depth interviews lasted 22–37 minutes, while key informant interviews took 19–46 minutes.

### Data analysis

The recorded interviews were transcribed verbatim in Amharic and conceptually transcribed into English. The translated data were then imported into Open Code 4.03 software. The investigator read and re-read the transcripts until intimately familiar with the content. Participants' views on the meaning of the events were considered during interpretation. A series of codes was developed, and line-by-line coding was carried out. Based on the relationships among the codes, sub-themes were developed and organized into main themes. These results were presented thematically under the main and sub-themes, and finally triangulated with the quantitative findings.

### Trustworthiness

The credibility, transferability, dependability, and conformability of the findings were ensured through various techniques. Triangulation of data was achieved by conducting in-depth interviews and engaging a diverse group of key informants. The interviewer established rapport with the interviewees by warmly greeting them and explaining the study's purpose to facilitate a comfortable atmosphere that promoted self-expression. Familiarity with participants' cultural and social backgrounds facilitated respect for their perspectives. Each interview began with general and light questions, transitioning to specific inquiries using verbal affirmations like 'aha' and maintaining eye contact to exhibit active listening and genuine

interest in their ideas. Probing questions were asked to obtain detailed information in response to participants' answers. At the conclusion of interviews, respondents were asked to verify if the summaries accurately captured their perspectives. Each transcript was cross-checked with audio recordings and field notes to maintain integrity. The researcher's professional background enhanced the study's credibility, including an MSc degree in pediatric nursing, current enrollment as a PhD candidate in public health, and experience conducting both quantitative and qualitative research.

### Ethical considerations

Ethical approval was obtained from the Bahir Dar University, College of Medicine and Health Sciences Institutional Review Board (IRB) (Protocol number: 703/2023). Additionally, a support letter was secured from the Amhara Public Health Institute. Relevant officials were informed hierarchically, and their permission letters were obtained. Prior to participation, informed consent was obtained from all mothers. Written informed consent was obtained from literate mothers. For illiterate mothers, voluntary participation was ensured by providing a detailed explanation of the study's purpose in a language they understood. Following their agreement to participate, a fingerprint was obtained. Confidentiality was maintained by excluding personal identifiers from the data collection questionnaire, and the data were kept in a locked board.

## Results

### Sociodemographic characteristics of the study participants

This study comprised 802 mothers (mean age, 29.6 ± 5.7 years) with infants aged 6–8 months. Almost all participants, 799 (99.6%), were Orthodox Christian by religion, and the majority, 778 (97%), were married. Most of study participants, 735 (91.7%), were engaged in farming. Slightly over half, 428 (53.4%), mothers had no formal education. Nearly four-fifths, 645 (80.4%), of mothers made joint household decisions with their husbands. In 491 households (61.2%), there were four to six residents, and nearly half, 400 (49.9%), had two children under five. Half of the infants, 404 (50.4%), were female. Additionally, 159 mothers (19.8%) were from the poorest households, while 161 (20.0%) were from the wealthiest households (Table 1).

### Maternal obstetric characteristics and infant health

Overall, 481 (60%) of the respondents were multiparous. Among the 733 mothers (91.4%) who attended antenatal checkups during the index pregnancy, more than half, 425 (58%), had four or more ANC visits. More than four-fifths of mothers, 663 (82.7%), had facility-based deliveries. More than half of the participants, 501 (62.5%), received PNC, and 411 (51.2%) mothers were counseled about CFPs during the postnatal period. Among 636 (79.3%) mothers with two or more livebirths, 396 (62.3%) delivered the index child after 34 months of the previous birth. Furthermore, 107 (13.3%) of infants experienced diarrhea within two weeks prior to the survey (Table 2).

### Maternal knowledge and attitudes toward ACFPs

Most respondents, 705 (87.9%), were aware of ACFPs. Among these, 540 (76.6%) mothers had received information from health professionals. Furthermore, 708 (88.3%) mothers demonstrated good knowledge of ACFPs, while 267 (33.3%) held positive attitudes toward it (Table 3).

### Food groups provided to infants during the previous day

Nearly all infants, 798 (99.5%), were breastfed and nearly three-fourths, 591 (73.7%), consumed food made from grains, roots and tubers during the previous day. Pulses, nuts, and seeds, consumed by 273 infants (34%), representing the third commonly consumed food group. In contrast, flesh foods (meat, fish, poultry, or organ meat), were the least consumed, reported for only 19 infants (2.4%) (Fig 2).

**Table 1. Sociodemographic characteristics of mothers with infants aged 6–8 months in West Gojjam Zone, Northwest Ethiopia 2023 (n = 802).**

| Variables | Category | Frequency | Percent (%) |
|---|---|---|---|
| Mother's age in years | <=24 | 149 | 18.6 |
| | 25-34 | 464 | 57.8 |
| | >=35 | 189 | 23.6 |
| Religion | Orthodox | 799 | 99.6 |
| | Others | 3 | 0.4 |
| Mother's education | No formal education | 428 | 53.4 |
| | Primary education | 295 | 36.8 |
| | Secondary/higher | 79 | 9.8 |
| Mother's occupation | Farmer | 735 | 91.7 |
| | Housewife | 54 | 6.7 |
| | Others[a] | 13 | 1.6 |
| Current marital status | Married | 778 | 97.0 |
| | Others[b] | 24 | 3.0 |
| Husband's education (778) | No formal education | 237 | 30.5 |
| | Primary education | 476 | 61.2 |
| | Secondary/higher | 65 | 8.3 |
| Decision-making on household resources | Husband | 112 | 14.0 |
| | Wife | 45 | 5.6 |
| | Jointly | 645 | 80.4 |
| **Variables** | **Category** | **Frequency** | **Percent (%)** |
| Family size | 1-3 | 168 | 21.0 |
| | 4-6 | 491 | 61.2 |
| | >=7 | 143 | 17.8 |
| Sex of infant | Male | 398 | 49.6 |
| | Female | 404 | 50.4 |
| Age of infant in months | 6 months | 296 | 36.9 |
| | 7 months | 277 | 34.5 |
| | 8 months | 229 | 28.6 |
| Number of children under five | One | 376 | 46.9 |
| | Two | 400 | 49.9 |
| | Three | 26 | 3.2 |
| Wealth index | Poorest | 159 | 19.8 |
| | Poor | 162 | 20.2 |
| | Medium | 160 | 20.0 |
| | Rich | 160 | 20.0 |
| | Richest | 161 | 20.0 |

a: Government employee, merchant, or student. b: Single, divorced, or widowed.

## Complementary feeding practices for infants aged 6–8 months

Approximately 80% of mothers, 639 (79.7%), introduced SSSF to their infants at the recommended 6–8 months, and two-thirds, 536 (66.8%), reported feeding their infants two or more times during the previous day. However, fewer mothers met the overall feeding standards, with only 77 (9.6%) demonstrating ACFPs for their infants. Similarly, 190 mothers (23.7%) had provided eggs and/or flesh foods the previous day, while 110 infants (13.7%) met the MDD criteria (Table 4).

**Table 2. Maternal obstetric characteristics and health of infants aged 6–8 months in West Gojjam Zone, Northwest Ethiopia 2023 (n = 802).**

| Variables | Category | Frequency | Percent (%) |
|---|---|---|---|
| Parity of mothers | Primipara (1) | 165 | 20.6 |
| | Multipara (2–4) | 481 | 60.0 |
| | Grand multipara (5+) | 156 | 19.4 |
| Antenatal care (ANC) attended | Yes | 733 | 91.4 |
| | No | 69 | 8.6 |
| Number of ANC attended (n = 733) | <=3 times | 308 | 42.0 |
| | >=4 times | 425 | 58.0 |
| Place of delivery | Home | 139 | 17.3 |
| | Health facility | 663 | 82.7 |
| Postnatal care (PNC) attended | Yes | 501 | 62.5 |
| | No | 301 | 37.5 |
| Number of PNC attended (n = 501) | 1-2 times | 372 | 74.3 |
| | >=3 times | 129 | 25.7 |
| Postnatal counseling on complementary feeding | Yes | 411 | 51.2 |
| | No | 391 | 48.8 |
| Birth order | 1st or 2nd | 366 | 45.6 |
| | 3rd or 4th | 436 | 54.4 |
| Birth interval (636) | <35 months | 240 | 37.7 |
| | >=35 months | 396 | 62.3 |
| **Variables** | **Category** | **Frequency** | **Percent (%)** |
| Diarrhea in 2 weeks preceding the survey | Yes | 107 | 13.3 |
| | No | 695 | 86.7 |
| Fever in 2 weeks preceding the survey | Yes | 70 | 8.7 |
| | No | 732 | 91.3 |
| Acute respiratory infection in 2 weeks preceding the survey | Yes | 67 | 8.4 |
| | No | 735 | 91.6 |

**Table 3. Knowledge and attitudes of mothers regarding appropriate complementary feeding practices in West Gojjam Zone, Northwest Ethiopia 2023 (n = 802).**

| Variables | Category | Frequency | Percent (%) |
|---|---|---|---|
| Ever heard about appropriate complementary feeding practices | Yes | 705 | 87.9 |
| | No | 97 | 12.1 |
| Source of information about appropriate complementary feeding practices (n = 705) | Friends/family | 44 | 6.2 |
| | Health professionals | 540 | 76.6 |
| | Media (Radio/TV) | 26 | 3.7 |
| | Others[a] | 95 | 13.5 |
| Mother's knowledge about appropriate complementary feeding practices | Good | 708 | 88.3 |
| | Average | 77 | 9.6 |
| | Poor | 17 | 2.1 |
| Mother's attitudes toward appropriate complementary feeding practices | Positive | 267 | 33.3 |
| | Negative | 535 | 66.7 |

a: Health professional and friends/family; health professional and friends/family and media.

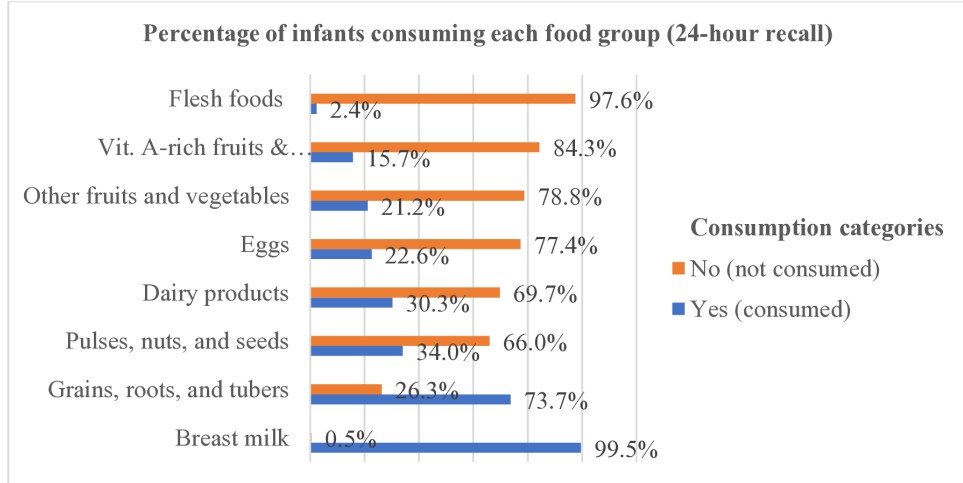

**Fig 2. Food groups provided for infants aged 6–8 months during the previous day (n = 802).**

## CFPs among mothers by IYCF indicators and infant age

Among 639 (79.7%) mothers who introduced SSSF to their infants (Table 4), 49.7% and 100% of infants aged 6 and 8 months, respectively, had consumed SSSF during the previous day. Of 536 (66.8%) mothers who met the MMF criteria for their infants the previous day (Table 4), feeding frequency increased with infant age: 33.1% at 6 months and 93.5% at 8 months. Among 77 mothers (9.6%) who met ACFPs for their infants, appropriate feeding also increased with infant age: 2.0% at 6 months and 17.9% at 8 months (Fig 3).

## Factors associated with ACFPs in mothers of 6–8-month-old infants

In bivariable analysis, infant age, postnatal counseling on CF, feeding animal-source foods (ASFs) on fasting days, wealth index, and perceived susceptibility were statistically associated with ACFPs. Variables with a p-value <0.25, including maternal occupation, household decision-making, birth order, and source of information about CFPs, were included in the multivariable analysis.

In the multivariable analysis, the independent predictors of ACFPs were infant age, postnatal counseling on CF, feeding ASFs on fasting days, wealth index, and perceived susceptibility. For each one-month increase in the infant's age, the odds of ACFPs among mothers increased threefold [AOR = 2.92, 95% CI: (1.99, 4.29)]. Mothers who received postnatal counseling on CF were 2.6 times [AOR = 2.64, 95% CI: (1.46, 4.75)] more likely to practice CF appropriately compared to those who did not. Mothers who fed ASFs to their infants on fasting days were also 2.6 times [AOR = 2.60, 95% CI: (1.20, 5.66)] more likely to practice CF appropriately than those who did not. Mothers from wealthier households had higher odds of practicing appropriate CF: rich households 3.1 times [AOR = 3.13, 95% CI: (1.32, 7.40)], and richest households 3.2 times [AOR = 3.16, 95% CI: (1.34, 7.49)] compared to poorest households. Additionally, mothers who perceived a high risk of infant undernutrition were 2.5 times [AOR = 2.45, 95% CI: (1.39, 4.31)] more likely to practice CF appropriately than their counterparts (Table 5).

## Barriers and facilitators of complementary feeding practices

Thematic analysis was conducted on the qualitative data, resulting in three main themes and five sub-themes. The three main themes include CFPs of mothers, barriers of ACFPs, and facilitators of CFPs.

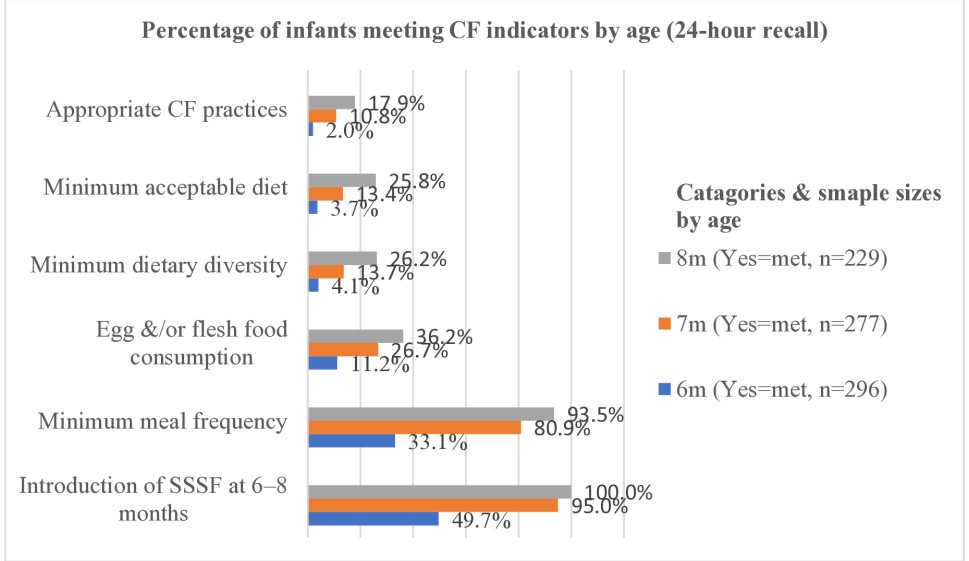

**Table 4. Complementary feeding practices among mothers with infants aged 6–8 months in West Gojjam Zone, Northwest Ethiopia, 2023 (n = 802).**

| Infant and young child feeding indicators | Categories | Proportion | 95% CI |
|---|---|---|---|
| Introduction of solid, semi-solid, or soft foods at 6–8 months | Yes | 79.7 | 76.7-82.4 |
|  | No | 20.3 | 17.6-23.3 |
| Feeding animal-source foods on fasting days | Yes | 72.1 | 68.8-75.2 |
|  | No | 27.9 | 24.8-31.2 |
| Minimum dietary diversity | Yes | 13.7 | 11.4-16.3 |
|  | No | 86.3 | 83.7-88;6 |
| Minimum meal frequency | Yes | 66.8 | 63.5-70.1 |
|  | No | 33.2 | 29.9-36.5 |
| Minimum acceptable diet | Yes | 13.3 | 11.1-15.9 |
|  | No | 86.7 | 84.1-88.9 |
| Egg and/or flesh food consumption | Yes | 23.7 | 20.8-26.8 |
|  | No | 76.3 | 73.2-79.2 |
| Appropriate complementary feeding practices | Yes | 9.6 | 7.7- 11.9 |
|  | No | 90.4 | 88.1-92.3 |

**Fig 3. Complementary feeding practices among mothers by IYCF indicators and infants' age (n = 802).** CF. Complementary feeding, SSSF. Solid, semi-solid, or soft foods.

## Theme 1: CFPs of mothers for their children

All interviewees reported poor child feeding practices in the study area, citing a lack of knowledge about diverse diets, reluctance toward CF and, a lack of awareness about the consequences of inappropriate complementary feeding practices.

A mother said, *"Because we are farmers, we do not know much about diverse diets. But when the child eats porridge and breastfeeds, I consider it appropriate food."*

**Table 5. Factors associated with appropriate complementary feeding practices among mothers with infants aged 6–8 months, West Gojjam Zone, Northwest Ethiopia, 2023.**

| Variables | Complementary feeding practices | | COR (95% CI) | AOR (95% CI) |
|---|---|---|---|---|
| | Appropriate n (%) | Inappropriate n (%) | | |
| **Mother's education** | | | | |
| No education | 40 (9.3) | 388 (90.7) | 1 | |
| Primary education | 29 (9.8) | 266 (90.2) | 1.058 (.640, 1.749) | |
| Secondary/higher | 8 (10.1) | 71 (89.9) | 1.093 (.491, 2.433) | |
| **Mother's occupation** | | | | |
| Farmer | 74 (10.1) | 661 (89.9) | 1 | |
| Housewife | 2 (3.7) | 52 (96.3) | .344 (.082, 1.439)* | |
| Others[a] | 1 (7.7) | 12 (92.3) | .744 (.095, 5.806) | |
| **Decision-making** | | | | |
| Husband | 7 (6.3) | 105 (93.7) | 1 | |
| Wife | 2 (4.4) | 43 (95.6) | .698 (.139, 3.494) | |
| Jointly | 68 (10.5) | 577 (89.5) | 1.768 (.790, 3.955)* | |
| **Family size** | | | | |
| 1-3 | 14 (8.3) | 154 (91.7) | .674 (.320, 1.420) | |
| 4-6 | 46 (9.4) | 445 (90.6) | .766 (.425, 1.383) | |
| >=7 | 17 (11.9) | 126 (88.1) | 1 | |
| **Sex of infant** | | | | |
| Male | 42 (10.6) | 356 (89.4) | 1.244 (.776, 1.993) | |
| Female | 35 (8.7) | 369 (91.3) | 1 | |
| **Age of infant** | | | 2.689 (1.925, 3.755)* | 2.924 (1.992, 4.294)** |
| Variables | Complementary feeding practices | | COR (95% CI) | AOR (95% CI) |
| | Appropriate n (%) | Inappropriate n (%) | | |
| **Birth order** | | | | |
| 1st or 2nd | 29 (7.9) | 337 (92.1) | .696 (.429, 1.128)* | |
| 3rd or 4th | 48 (11.0) | 388 (89.0) | 1 | |
| **No of <5 children** | | | | |
| 1 | 25 (6.6) | 351 (93.4) | .546 (.153, 1.944) | |
| 2 | 49 (12.3) | 351 (87.7) | 1.070 (.310, 3.697) | |
| 3 | 3 (11.5) | 23 (88.5) | 1 | |
| **ANC attended** | | | | |
| No | 5 (7.2) | 64 (92.8) | 1 | |
| Yes | 72 (9.8) | 661 (90.2) | 1.394 (.544, 3.577) | |
| **Postnatal CFC** | | | | |
| No | 26 (3.2) | 365 (96.8) | 1 | 1 |
| Yes | 51 (12.4) | 360 (87.6) | 1.989 (1.213, 3.260)* | 2.635 (1.46, 4.752)** |
| **Information source** | | | | |
| Friends/family | 1 (2.3) | 43 (97.7) | 1 | |
| Health professional | 52 (9.6) | 488 (90.4) | 4.582 (.618, 33.963)* | |
| Media (Radio/TV) | 2 (7.7) | 24 (92.3) | 3.583 (.309, 41.602) | |
| Others[b] | 12 (12.6) | 83 (87.4) | 6.217 (.782, 49.414) | |
| **Feeding ASFs on fasting days** | | | | |
| No | 10 (4.5) | 214 (95.5) | 1 | 1 |
| Yes | 67 (11.6) | 511 (88.4) | 2.806 (1.417, 5.557)* | 2.602 (1.197, 5.655)** |

*(Continued)*

**Table 5.** (Continued)

| Variables | Complementary feeding practices | | COR (95% CI) | AOR (95% CI) |
|---|---|---|---|---|
| | Appropriate n (%) | Inappropriate n (%) | | |
| Wealth index | | | | |
| Poorest | 12 (7.5) | 147 (92.5) | 1 | 1 |
| Poor | 9 (5.6) | 153 (94.4) | .721 (.295, 1.761) | .723 (.252, 2.076) |
| Medium | 8 (5.0) | 152 (95.0) | .645 (.256, 1.622) | .839 (.304, 2.314) |
| Rich | 21 (13.1) | 139 (86.9) | 1.851 (.878, 3.903)* | 3.127 (1.321, 7.404)** |
| Richest | 27 (16.8) | 134 (83.2) | 2.468 (1.202, 5.067)* | 3.164 (1.337, 7.490)** |
| Knowledge on ACFP | | | | |
| Good | 70 (9.9) | 638 (90.1) | 1.755 (.229, 13.437) | |
| Average | 6 (7.8) | 71 (92.2) | 1.352 (.152, 12.026) | |
| Poor | 1 (5.9) | 16 (94.1) | 1 | |
| Attitudes on ACFPs | | | | |
| Negative | 55 (10.3) | 480 (89.7) | 1 | |
| Positive | 22 (8.2) | 245 (91.8) | .784 (.467, 1.315) | |
| P-susceptibility | | | | |
| Low | 43 (7.5) | 528 (92.5) | 1 | 1 |
| High | 34 (14.7) | 197 (85.3) | 2.119 (1.313, 3.420)* | 2.448 (1.390, 4.311)** |
| P-severity | | | | |
| Low | 56 (10.4) | 485 (89.6) | 1 | |
| High | 21 (8.0) | 240 (92.0) | .758 (.448, 1.281) | |
| P-benefits | | | | |
| Low | 50 (10.0) | 451 (90.0) | 1 | |
| High | 27 (9.0) | 274 (91.0) | .889 (.544, 1.453) | |
| Variables | Complementary feeding practices | | COR (95% CI) | AOR (95% CI) |
| | Appropriate n (%) | Inappropriate n (%) | | |
| P-barriers | | | | |
| Low | 41 (9.8) | 379 (90.2) | 1.040 (.649, 1.665) | |
| High | 36 (9.4) | 346 (90.6) | 1 | |

*Candidate variables for multivariable analysis, **Variables with a p-value <0.05.

a: Government employee, merchant, student, b: Health professional and friends/family; health professional, friends/family, and media, ACFPs: Appropriate complementary feeding practices, ANC: Antenatal care, AOR: Adjusted odds ratio, ASFs: Animal-source foods, CFC: Complementary feeding counselling, CI: Confidence interval, COR: Crude odds ratio, PNC: Postnatal care, P-: Perceived.

Another key informant also explained, *"…I think it is a problem of not understand well how much the actual problem [undernutrition] results from a lack of CFs… There is also the attitude that, "We have also grown up like this. When pressure is applied to prepare complementary foods, they [mothers] often say, "The child is unwilling to eat." It is searching for reasons."*

Furthermore, all participants noted that feeding ASFs, particularly meat to children aged 6–23 months is uncommon or done irregularly. This is often because parents assume that meat cannot be easily digested, has no benefit for children or is available only during the annual festivals.

*"There is no habit of thinking that meat will benefit them [children] apart from the family. As it is often known, meat is available at the time of the festivals … for example, meat is available during Christmas, Easter, and other annual*

*holidays. At other times, there is often no meat. Even when it is available, the priority is given to adults rather than children."*

**Theme 2: Barriers to complementary feeding practices**

From the in-depth and key informant interview data, three sub-themes emerged: excessive workload, misconceptions, and economy/poverty. These sub-themes illustrate the barriers mothers faced in implementing ACFPs for their children.

**Sub-theme 1: Excessive workload.** Most participants reported that excessive workload and a lack of time hinder ACFPs. They cited household chores, family management, and farm work as barriers that make it difficult to prepare and provide the recommended foods.

*"Number of jobs and lack of time. First, we are farmers. A farmer has no rest. We plough the farmland; then there are seedlings. Once planted and grow, there is a weeding, followed by harvesting. We also have cows, but there is no child [elder child] to help me. I am the one who rises and falls [I do everything] with no rest. I run together the cattle, fetch water from the well, and water the cows."*

**Sub-theme 2: Misconceptions.** The participants noted misconceptions regarding CFPs. Most key informants emphasized that mothers often delay introducing CFs after six months because they perceive infants' stools to be copious and smelly, which makes them feel ashamed to socialize with others.

*"The reasons they [mothers] give often confuses you. They make you laugh. What do you think they will say? What if I start complementary food at six months? What about their feces? They say their stools are abundant and smelly. From now on? What if we sit in front of people? You will find mothers who say this (Laughs)."*

All in-depth interviewees noted that giving undiluted cow's milk to young children is not recommended, as it is considered too heavy for their digestive systems. They noted that because milk is not separated from butter, a significant amount of water is added to dilute it during boiling.

A mother described, *"I will boil one hand [cup] of milk [cow's milk] with two hands [cups] of water. This is because I was told that… the milk (without water) is too thick and uncomfortable for the child…"*

Most in-depth interviewees and key informants reported that providing ASFs for children on fasting days is uncommon in the study area. It is believed that offering ASFs on fasting days would make mothers appear as if they are not fasting (*tsome yasegedefale*).

*"There are many mothers who say "I do not prepare animal-source foods for the child on fasting days. Most people do not give meat or eggs during the fasting time. The problem is that, why do I smell it while the child is eating? Either I do not taste it? or they assume, I smell it and I will be considered as not fasting (tsome egedefalehu). They say the utensils will be contaminated, and I might forget and touch the adult utensils..."*

Furthermore, most participants described that red organ meats, such as liver, kidney, and heart, which are rich in iron, are not given to children aged 6–23 months because they believe these types of meat are intended for adults. A mother said, *"Regarding this [organ meat] …I do not know (laughs). I am not used to feeding these [liver, kidney, and heart] to the child. I never even thought to give these to the child."*

Another key informant also added, *"There is a need to give priority to children and give these types of meat (liver, heart, and kidney). Many mothers often have no idea that their children need these types of meat. There is a gap [knowledge gap] in this case. Particularly, there is a lack of understanding that liver, heart, and kidneys are important for children."*

**Sub-theme 3: Economy (poverty).**  Most participants mentioned that monotonous food is frequently given to children due to families' inability to provide diverse diets or the high cost of doing so.

One mother explained, *"Mothers around me [here] feed their children like I do. Like me. We feed children what we have. We simply feed them porridge, or atmit (soup-like fluid), or shiro [fermented flatbread with legume-based dish]. No milk, meat, or vegetables."*

Another key informant also emphasized, *"In terms of price, some grains and legumes are now very expensive. If they [mothers] do not have it at home, for example, peas, beans, grains, or oil, they are all expensive. There is no such thing as cheap, it may be difficult for them [mothers] to prepare that [diverse meals]."*

In contrast, most key informants reported that the community frequently sell nutrient-dense foods like eggs and butter due to their high cost, while simultaneously purchasing cheaper, less nutritious food items to meet household needs.

*"…if there are items like eggs that are good for children, they [families] take them to the market for sell and to buy other simpler [less nutritious] food items. They do not understand the nutritional differences between what they sell and buy. They focus on the profit they make. The practice of feeding nutritious food has not yet developed, and we still have a lot to do."*

## Theme 3: Facilitators of complementary feeding practices

The facilitator sub-themes of CFPs were categorized into two main areas: community support for infant and young child feeding (IYCF) practices and the network of the health care system.

**Sub-theme 1: Community support for IYCF practices.**  All participants reported that relatives and neighbors celebrate the new mother after she gives birth, often saying, *"Enkuan Mareyam marechish."* They offer psychological support and advise her to consume nutritious foods and drinks (e.g., cow's milk), exclusively breastfeed for six months, and introduce CFs timely.

*"It is always a joy when a child comes into this world. The family and the surrounding community are happy when a mother gives birth. …This means she [mother] will be advised on what to do after delivery. She needs to be fed well, she is informed about when to start complementary feeding, and she is monitored. This is the convenient situation for CFPs."*

**Sub-theme 2: Network of the healthcare system.**  Nearly all participants described the hierarchical network between healthcare facilities and the community. This linkage enables the community to access information on IYCF, receive lessons and, at times, mothers attend complementary food cooking demonstrations at the *kebele* level.

A mother explained, *"The first place where we get information is when we come here [health center] for antenatal care. There is advice at that time. …We will be monitored, receive lessons, and do our best to implement the lessons we have been taught. When we give birth here, the health professionals teach us to breastfeed exclusively for the first six months, and then to introduce CFs."*

## Discussion

Child undernutrition begins to increase at 6 months [26] due to inadequate nutrition from breast milk and inappropriate complementary feeding practices (CFPs) [43]. In this study, the prevalence of appropriate complementary feeding practices (ACFPs) among mothers with infants aged 6–8 months was only 9.6% [95% CI: (7.7, 11.9)]. Poor CFPs among mothers were also reported by qualitative study participants, attributing to a lack of knowledge about diverse diets, reluctance toward complementary feeding (CF) and a lack of awareness about the consequences of inappropriate CFPs.

This finding is consistent with studies conducted in Ethiopia, including those from Arsi Negele district (9.5%), Horro district (9.91%), and a study based on the 2019 Ethiopia Mini Demographic Health Survey (EMDHS) (9.76%) [10,17,18]. However, the 9.6% prevalence reported in this study is lower than those observed in Shashemene (30%) and Debre Tabor Hospital (37.2%), Ethiopia [15,16]. This discrepancy might be attributed to differences in study settings, as this study was conducted in rural areas with limited access to CF counseling and health services, when compared with urban areas. Our study also covered multiple districts, while the previous studies focused on a single town or district population.

Furthermore, the 9.6% prevalence of ACFPs is lower than findings from northern Ghana (14.3%), an urban community in Lagos State, Nigeria (47%) and the Bhimphedi rural municipality of Nepal (25%) [44–46]. The differences might stem from socioeconomic and cultural factors. This study's maternal education level was lower than in the comparison studies, which could affect access to information about ACFPs. Additionally, the variation might result from differences in study populations. This study included infants aged 6–8 months, while the comparison studies included children up to 23 months, suggesting that ACFP may improve as children grow older [16,18,45,47].

The very low prevalence rate of ACFPs identified in this study carries important practical and policy implications. From a practical perspective, these findings highlight the need for targeted interventions to improve ACFPs among mothers with infants. For policymakers, the results provide valuable evidence to strengthen infant and young child feeding (IYCF) programs and policies. Improving ACFPs would directly contribute to achieving Ethiopian's Seqota Declaration of Zero Hunger, which aims to "end stunting in children under two" [25]. Furthermore, these efforts align with global commitments, particularly Sustainable Development Goal target 2.2, which seeks to "end all forms of malnutrition by 2030" [48].

The most consumed food group by infants in this study area was grains, roots, and tubers, accounting for 73.7% [95% CI: (70.5, 76.7)]. In the qualitative study, all participants reported that grains, roots, and tubers, along with legumes and nuts, were the common food groups given to children in the form of porridge or *atmit* (soup-like fluid). Similarly, many studies conducted in different districts of Ethiopia have noted that grains, roots, and tubers are the commonly consumed food group among infants aged 6–11 months. For instance, 82.8% of infants in Shashemene, 97.8% in Horro, 87.7% in Damot Sore, and 94.9% in the Lasta district consumed food made from these staples [10,14,16,49]. This suggests that cereal-based food items are the most common form in which mothers introduce complementary foods (CFs).

In this study, 79.7% [95% CI: (76.7, 82.4)] of mothers introduced CFs for infants aged 6–8 months at the time recommended by the WHO [2]. This finding is close to the WHO cut-off, which indicates that over 80% of infants should start CFs at 6–8 months. The 79.7% result is comparable to findings from a study in Bhimphedi rural municipality of Nepal (82%) [44], and Malawi (83.2%) [9]. However, it is higher than the national figure (69%) reported in the 2019 EMDHS [3].

The lower national percentage may reflect the inclusion of diverse populations with varying sociocultural backgrounds and CFPs in the EDHS data. This 79.7% prevalence also exceeds the results from Southwestern Nigeria (50.9%) [28], as well as those from Iseyin, Nigeria (72.3%) [47]. This might be because health extension workers in each *kebele* promote the timely introduction of CFs [50], helping mothers initiate feeding at the appropriate time.

The prevalence rate of minimum dietary diversity (MDD) in this study was only 13.7% [95% CI: (11.4, 16.3)]. Most participants in the qualitative study noted that monotonous food, such as porridge, *atmit* (soup-like fluid), or fermented flatbread with legume-based dish, were commonly provided to children. This is due to families' inability to provide a diverse diet or the high cost of food items. The 13.7% prevalence of MDD is consistent with findings from other Ethiopian studies, including two based on the 2019 EMDHS: 11.7% (20) and 14% (24), as well as studies conducted in Horro District (15.8%) [10] and Damot Sore District (16.5%) [49]. This prevalence is also comparable to a study finding from Southwestern Nigeria (14.5%) [28].

However, this 13.7% prevalence is lower than rates reported in Shashemene, Ethiopia (42.5%), northern Ghana (35.3%), and Malawi (22.7%) [9,16,45]. These disparities likely reflect regional variations in urban-rural residence, feeding culture, and socioeconomic status. Additionally, the current study's lower prevalence might result from differences in considering breast milk as part of the eight-food group, compared with calculating MDD based on the seven food groups.

Evidence indicates that the prevalence of MDD based on the eight food groups is lower than when calculated using the seven food group indicators outlined in the 2008 guidelines, across all countries in the Eastern and Southern Africa Region and in Bangladesh [21,22].

Infants who met the minimum meal frequency (MMF) accounted for 66.8% [95% CI: (63.5, 70.1)]. This result is congruent with a study conducted in Arsi Negele district, Ethiopia (67.3%) [18] and in Bhimphedi rural municipality of Nepal (71%) [44]. However, it is lower than findings from Debrelibanos district, Ethiopia (79.2%) [27], Damot sore district, Ethiopia (94.5%) [49], and Nepal (82%) [51]. The lower MMF prevalence observed in this study might be attributed to population differences: we focused exclusively on infants aged 6–8 months, whereas comparison studies included children up to 23 months, an age group that typically exhibits higher meal frequency due to greater dietary requirements.

The prevalence of infants meeting the minimum acceptable diet (MAD) in this study was 13.3% [95% CI: (11.1, 15.9)]. This result is comparable to the findings of a national figure (11%) [3], as well as studies from Horro District, Ethiopia (10.5%), the 2019 EMDHS data (11.5%), and Iseyin, Nigeria (14.9%) [10,43,47]. However, the 13.3% prevalence reported in this study exceeds the Amhara regional estimate of 6% [3]. This disparity may stem from the broader geographic coverage of the EMDHS, which encompassed populations with diverse sociocultural contexts and child feeding practices. Conversely, the findings of this study were lower than those reported in northern Ghana (25.2%) and Bhimphedi rural municipality of Nepal (27%) [44,45]. These variations could reflect differences in cultural practices, socioeconomic conditions, and policy implementations across regions. The lower MAD proportion in this study suggests most infants either received meals below the recommended frequency (<2 times/day) or lacked dietary diversity (<5 food groups) (5).

In the present study, only 23.7% [95% CI: (20.8, 26.8)] of mothers fed egg and/or flesh food for their infants during the previous day. All participants in the qualitative study reported that animal source foods (ASFs), particularly meat, were seldom fed to infants. They attributed this practice to three main factors: the belief that meat is difficult for infants to digest, the perception that it offers no nutritional benefits, or only available during annual festivals. Furthermore, most key informants reported that the community frequently sell nutrient-dense foods like eggs and butter due to their high cost, while purchasing cheaper, less nutritious items to meet household needs.

Similarly, studies have reported that flesh foods are the least commonly consumed among infants aged 6–8 months [47] and those aged 6–11 months [10,14,16]. Another study revealed that only 36% of children aged 6–36 months from agrarian families were fed ASFs in the 24 hours preceding the survey [52]. This suggests that many mothers feed their children monotonous diets, despite ASFs providing high-quality proteins and essential micronutrients often lacking in plant-based options, especially for young children with higher nutritional requirements [53].

Infant's age was found to be a predictor variable; for each one-month increase in the infant's age, the odds of ACFPs among mothers increased threefold. All in-depth interview mothers expressed the misconception that infants lack digestive capacity for solid foods in early months. Thus, a significant amount of water is added to dilute cow's milk during boiling. Many studies conducted in Ethiopia [10,16–18], northern Ghana [45], and Iseyin Nigeria [47] have consistently identified older child age (12–23 months) as a predictor of ACFPs, when compared with younger infants (6–11 months).

These findings suggest that most mothers' CFPs are inadequate during the early months when CF typically begins. This presents an opportunity for healthcare providers to promote proper infant feeding practices by addressing these misconceptions through effective counseling.

This study found that mothers who received postnatal counseling on CF were more likely to practice appropriate infant feeding compared to those who did not. All respondents in the qualitative study noted that relatives and neighbors visit a new mother and encourage to consume nutritious foods, exclusively breastfeed for six months, and timely introduce CFs. Supporting this finding, studies from different districts in Ethiopia: Lasta district [14]. Bensa district [54], Damot Weydie district [19], as well as a study conducted in Sub-Saharan Africa using Demographic Health Survey data from 19 countries [55] showed that mothers who received postnatal care services were more likely to practice CF appropriately than mothers who did not. This suggests postnatal feeding counseling and support are essential to promote appropriate infant

feeding practices, which in turn improve maternal and child health outcomes. Similarly, both lack of postnatal check-ups and health education on CF predicted inappropriate CFPs [30]. This might be attributed to insufficient counseling when postnatal visits are missed.

Mothers who fed their infants ASFs on fasting days had increased odds of practicing appropriate CF compared to those who did not. However, most qualitative study respondents reported that providing ASFs for children on fasting days was uncommon in the area. They attributed this practice to the belief that doing so would appear as if mothers were not fasting (*tsome yasegedefale*). This finding aligns with an Alive and Thrive report from Amhara region of Ethiopia, which noted that mothers often avoid providing ASFs on fasting days due to concerns about neighbors' disapproval, utensils contamination, and the smell of butter interfering with their own or others' fasts [56].

Studies conducted in three districts of the Amhara region, Ethiopia revealed consistent patterns in child feeding during fasting periods: In Debrebirhan, flesh food consumption was significantly higher on non-fasting days [57], while in Gondar City, children fed ASFs during fasts were more likely to meet MAD [58]. In Dejen District, mothers avoiding ASFs on fasting days had lower dietary diversity scores [59]. These practices appear to be influenced by cultural norms, including mothers' adherence to fasting rules and the avoidance of separate utensils during fasting periods.

The odds of ACFPs were higher among mothers from wealthier households compared with those from the poorest households. In the qualitative interviews, two in-depth interview mothers and most key informants identified financial constraints as a barrier to mothers' adoption of ACFPs for their children. This result is consistent with previous research from Shashemene, Ethiopia, a study using the 2019 EMDHS, and a study in Sub-Saharan Africa [16,17,60]. Similarly, a study in Dessie City, Ethiopia [30] identified poor economic status as a predictor of inappropriate CFPs, while Pakistan data confirmed household wealth as a predictor of meeting MDD [61]. These findings collectively demonstrate that household wealth strongly influences families' capacity to meet diverse dietary requirements, ultimately affecting their adoption of ACFPs.

Mothers who perceived their infants to be at high risk of undernutrition had higher odds of practicing appropriate CF compared to those who perceived at low risk. This finding aligns with research from Ghana showed that the belief in breastmilk's inability to satisfy hunger was associated with perceived readiness for CF [62]. Similarly, a study conducted at the East Surabaya Health Center in Indonesia found that mothers' knowledge of stunting prevention is influenced by their perceived susceptibility [63]. These findings suggest that mothers' adoption of appropriate child feeding practices is influenced by their perception of nutritional risks.

This study combined quantitative and qualitative approaches to examine complementary feeding practices. Only 9.6% of mothers with infants aged 6–8 months adhered to ACFPs. The infant's age, receipt of postnatal counseling on CF, provision of ASFs on fasting days, higher household wealth, and maternal perception of undernutrition risks were significantly associated with ACFPs. Qualitative findings strengthened these results, highlighting how cultural norms, religious practices, limited understanding of diverse diets, and financial constraints contributed feeding behaviors. While the quantitative analysis identified statistical associations, qualitative interviews revealed how societal expectations, time constraints, and traditional beliefs further hinder adherence to ACFPs. These findings underscore the need for multifaceted, context-specific interventions that address the identified barriers and social and cultural norms.

This study's strength lies in its coverage of a large geographic area in West Gojjam Zone and its use of a representative sample size. We assessed the prevalence of MDD, MAD, and ACFPs using the revised WHO IYCF indicators. Furthermore, we incorporated qualitative data to explore barriers and facilitators of CFPs. This study also acknowledges potential biases. Mothers' self-reports on their infants' food consumption and feeding frequency, may lead to over- or underestimation. Using a 24-hour dietary recall may introduce recall bias, as mothers may not accurately remember their infants' dietary intake. Focusing on rural areas might limit the findings' applicability to urban populations. Social desirability bias in qualitative interviews may lead participants to give answers they believe are more acceptable. Although the

updated WHO IYCF indicators were used to assess MDD, MAD, and ACFPs, earlier findings based on the 2008 indicators were cited due to limited evidence supporting the new ones. These limitations should be considered when interpreting the results.

## Conclusion

In this study, the prevalence of appropriate complementary feeding practices among mothers with infants aged 6–8 months was very low. A month increase in the infant's age, postnatal counseling on CF, provision of ASFs on fasting days, wealth index, and perceived susceptibility were predictors of ACFPs. Moreover, excessive workload, cultural misconceptions, and economy (poverty) were barriers to these practices. Therefore, healthcare providers should strengthen postnatal counseling on CF and promote provision of age-appropriate ASFs on fasting days. Improving households' economic status and mothers' understanding of the risks associated with inappropriate CFPs is vital. Collaboration among stakeholders, including women's affairs offices and religious leaders, can help reduce mothers' workload and address cultural misconceptions about CFPs.

## Supporting information

**S1 Appendix. Interview guides (English and Amharic versions).**
(DOCX)

**S1 Dataset. Quantitative minimal dataset.**
(SAV)

**S2 Dataset. Qualitative minimal dataset.**
(PDF)

## Acknowledgments

We would like to thank all the study participants for their willingness. We would like to acknowledge the School of Public Health, College of Medicine and Health Sciences, Bahir Dar University for giving us the chance to do this research project. We would like to extend our thanks to the library workers of the College of Medicine and Health Sciences for their kind cooperation.

## Author contributions

**Conceptualization:** Shiferaw Birhanu, Getu Degu Alene, Yeshalem Mulugeta Demilew.

**Data curation:** Shiferaw Birhanu, Getu Degu Alene, Yeshalem Mulugeta Demilew.

**Formal analysis:** Shiferaw Birhanu, Getu Degu Alene, Yeshalem Mulugeta Demilew.

**Investigation:** Shiferaw Birhanu, Getu Degu Alene, Yeshalem Mulugeta Demilew.

**Methodology:** Shiferaw Birhanu, Getu Degu Alene, Yeshalem Mulugeta Demilew.

**Project administration:** Shiferaw Birhanu, Getu Degu Alene, Yeshalem Mulugeta Demilew.

**Supervision:** Shiferaw Birhanu, Getu Degu Alene, Yeshalem Mulugeta Demilew.

**Validation:** Shiferaw Birhanu, Getu Degu Alene, Yeshalem Mulugeta Demilew.

**Visualization:** Shiferaw Birhanu, Getu Degu Alene, Yeshalem Mulugeta Demilew.

**Writing – original draft:** Shiferaw Birhanu.

**Writing – review & editing:** Shiferaw Birhanu, Getu Degu Alene, Yeshalem Mulugeta Demilew.

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
