## [Decision Letter · Decision Letter 0]

28 Mar 2025

Dear Dr. Birhanu,

Thank you for submitting your manuscript to PLOS ONE. After careful consideration, we feel that it has merit but does not fully meet PLOS ONE’s publication criteria as it currently stands. Therefore, we invite you to submit a revised version of the manuscript that addresses the points raised during the review process.

We look forward to receiving your revised manuscript.

Kind regards,

Omnia Samir El Seifi, M.D., Ph.D.

Academic Editor

PLOS ONE

Journal Requirements:

2. We note that your Data Availability Statement is currently as follows: All relevant data are within the manuscript and in Supporting Information files.

Reviewers' comments:

Reviewer's Responses to Questions

**Comments to the Author**

1. Is the manuscript technically sound, and do the data support the conclusions?

Reviewer #1: Yes

Reviewer #2: Yes

Reviewer #3: Partly

2. Has the statistical analysis been performed appropriately and rigorously?

Reviewer #1: Yes

Reviewer #2: Yes

Reviewer #3: Yes

3. Have the authors made all data underlying the findings in their manuscript fully available?

Reviewer #1: Yes

Reviewer #2: Yes

Reviewer #3: No

4. Is the manuscript presented in an intelligible fashion and written in standard English?

Reviewer #1: Yes

Reviewer #2: Yes

Reviewer #3: Yes

Reviewer #1: The study provides valuable insights into the barriers and facilitators of appropriate complementary feeding practices among mothers with infants aged 6–8 months in rural Ethiopia. The rigorous methodology, including the mixed-methods approach, and the use of updated WHO indicators are strengths of the study. The findings are well-supported by the data and make a significant contribution to public health research in Ethiopia, particularly in improving infant and young child nutrition.

With some minor revisions, particularly in methodological clarity and presentation, this article would be suitable for publication.

Reviewer #2: Summary

This paper addresses an important issue related to appropriate complementary feeding practices among mothers with infants aged 6–8 months in Northwest Ethiopia. The study is valuable as it employs a mixed-methods approach, providing both quantitative and qualitative insights into the factors influencing complementary feeding practices. The findings have significant public health implications, particularly for improving infant nutrition in similar settings.

However, there are some corrections needed. The following comments highlight specific aspects that need attention.

Study Population

• The sentence in line 159 lacks clarity. Consider rephrasing for better readability.

Methods

• If possible, include a flowchart to illustrate the steps involved in data collection.

Independent Variables

• List all the independent variables together.

• Lines 381 and 396: Revise for clarity.

Results

• If possible, use charts or plots to visually represent key findings of multivariate analysis.

Themes

• Theme 1 and Theme 2: Review the statements from respondents for clarity. Some sentences require rewording to enhance readability and ensure accurate interpretation.

Discussion

• When comparing findings related to a specific variable with previous research, discuss them within their context. Instead of using "This study" to refer to a result, consider specifying the exact figure or result for clarity.

For example, lines 575 and 611–612 would benefit from such specificity.

There are a few grammatical inconsistencies. A thorough proofreading of the manuscript is recommended to ensure clarity and coherence.

Recommended course of action: Request Revision

Reviewer #3: Thank you for the opportunity to review this important and well-structured manuscript. The study addresses a critical public health issue through a thoughtful mixed-methods approach. I have provided detailed comments and suggestions for improvement in the attached reviewer report, including feedback on the methodology, integration of qualitative and quantitative findings, and clarity of presentation. I hope these suggestions are helpful in strengthening your work.

**Do you want your identity to be public for this peer review?** For information about this choice, including consent withdrawal, please see our Privacy Policy

Reviewer #1: No

Reviewer #2: No

Reviewer #3: **Yes: ** Muhammad Aasim

---

## [Author Response · Author response to Decision Letter 1]

10 May 2025

Response to reviewers

We sincerely appreciate your time and valuable feedback on our manuscript. Below, we have addressed each of your comments in detail. Please note that the page and line numbers in our responses correspond to those in the clean version submitted as “Manuscript.”

Editor

Journal Requirements: When submitting your revision, we need you to address these additional requirements.

Comment: 1. Please ensure that your manuscript meets PLOS ONE's style requirements, including those for file naming.

Response: Thank you for your feedback. We have made efforts to ensure that our manuscript meets the style requirements of PLOS ONE.

Comment: 2. We note that your Data Availability Statement is currently as follows: All relevant data are within the manuscript and in Supporting Information files.

Please confirm at this time whether your submission contains all raw data required to replicate the results of your study. Authors must share the “minimal data set” for their submission. For example, authors should submit the following data:

Response: Thank you for your observation regarding the Data Availability Statement. We attached minimal datasets for both the quantitative and qualitative study as supplementary files.

Comment: 3. When completing the data availability statement of the submission form, you indicated that you will make your data available on acceptance. We strongly recommend all authors decide on a data sharing plan before acceptance, as the process can be lengthy and hold up publication timelines. Please note that, though access restrictions are acceptable now, your entire data will need to be made freely accessible if your manuscript is accepted for publication. This policy applies to all data except where public deposition would breach compliance with the protocol approved by your research ethics board. If you are unable to adhere to our open data policy, please kindly revise your statement to explain your reasoning and we will seek the editor's input on an exemption. Please be assured that, once you have provided your new statement, the assessment of your exemption will not hold up the peer review process.

Response: Thank you for your feedback regarding sharing the data. We attached our minimal datasets for both the quantitative and qualitative study as supplementary files.

Comment: 4. Please include your full ethics statement in the ‘Methods’ section of your manuscript file. In your statement, please include the full name of the IRB or ethics committee who approved or waived your study, as well as whether or not you obtained informed written or verbal consent.

Response: Thank you for your comment. We have included our complete ethics statement, including the full name of the IRB and details about obtaining informed written consent or fingerprints, in the 'Methods' section (please refer to page 17, 361-370).

Reviewer 1

Comment: • The article is generally well-organized, with clear sections outlining the background, methodology, results, and discussion. Tables and figures are used effectively to present key findings (e.g., Tables 4 and 5 present a comprehensive breakdown of complementary feeding practices and factors associated with ACFP).

• However, the manuscript could benefit from more detailed descriptions of some of the statistical procedures, such as the generalized linear mixed model, as this might not be familiar to all readers.

Response: We are grateful for your thoughtful comment. We have tried to elaborate the generalized linear mixed model in detail to make it familiar to readers (kindly refer to page 14, lines 292-300).

Comment: The inclusion of more details on how the wealth index was calculated would enhance transparency in the methods.

Response: We have provided a detailed explanation of the wealth index calculation to enhance the transparency of our methods (please refer to pages 11 & 12, lines 240-257).

Comment: • The article generally adheres to standard reporting guidelines, including those for ethical conduct, informed consent, and transparency in data analysis.

However, further clarification on the data availability statement and adherence to data-sharing community standards (e.g., deposition in a public repository) could strengthen compliance with open data practices.

Response: Thank you for your suggestion. We attached minimal datasets for both the quantitative and qualitative study as supplementary files.

Comment: • The manuscript is written in clear, standard English with minimal grammatical errors. However, there are occasional awkward phrasings (e.g., "flesh foods" could be more clearly stated as "animal-source foods"). These minor issues do not significantly detract from the overall clarity but could be addressed to improve readability.

Response: We appreciate your suggestions for improving clarity. We have corrected grammatical errors throughout the manuscript. The phrase, “flesh foods” is used to specify the prevalence of meat consumption within the eight food groups considered for computing minimum dietary diversity. Using “animal source foods” may not clearly indicate this proportion, as “animal source foods’ include dairy products (milk, yogurt, and cheese), eggs, and flesh foods (please refer page 24, lines 456-460).

However, “animal-source foods” is used to describe whether mothers provided dairy products, eggs, or flesh foods to their children on fasting days (please refer page 42, lines 780-782).

Comment: • The manuscript could be improved by breaking up longer paragraphs, particularly in the discussion section, to enhance readability and clarity.

Response: Thank you for your feedback. We have improved the readability of the manuscript by breaking up long paragraphs in both the introduction and discussion sections.

Reviewer 2

Comment: Study Population

• The sentence in line 159 lacks clarity. Consider rephrasing for better readability.

Response: Thank you for your comment. The sentence in line 159 has been rephrased in the main document for clarity (page 8, line 159.

Comment: Methods

• If possible, include a flowchart to illustrate the steps involved in data collection.

Response: Thank you for your suggestion. Steps of data collection is illustrated using a flowchart for clarity. It is available as Fig 1 (kindly refer to page 11, lines 226-227).

Comment: Independent Variables

• List all the independent variables together.

Response: Thank you for your comment. All independent variables are listed together (please refer to page 11, line 232-238).

Comment: • Lines 381 and 396: Revise for clarity.

Response: Thank you for your recommendation. Both lines 381 and 396 have been corrected in the main document for clarity (please refer to pages 23, line 440-442) and page 24, lines 456-460).

Comment: Results

• If possible, use charts or plots to visually represent key findings of multivariate analysis.

Response: We appreciate your suggestion to enhance the visual representation of our multivariable analysis results. Currently, Table 5 presents the key multivariable analysis findings in a detailed format which includes all relevant statistical parameters (odds ratios, confidence intervals, and p-values). While we recognize the value of charts/plots for visualizing data, we found that the tabular format better preserves the precision needed for interpreting these key findings. However, we would be happy to add a supplementary figure if deemed essential (kindly refer to pages 28-231).

Comment: Themes

• Theme 1 and Theme 2: Review the statements from respondents for clarity. Some sentences require rewording to enhance readability and ensure accurate interpretation.

Response: Thank you for your feedback. Some words, phrases and sentences are reworded to enhance readability and ensure accurate interpretation for both theme 1 and 2 (please refer to pages 31-35).

Comment: Discussion

• When comparing findings related to a specific variable with previous research, discuss them within their context. Instead of using "This study" to refer to a result, consider specifying the exact figure or result for clarity. For example, lines 575 and 611–612 would benefit from such specificity.

Response: Thank you for your insightful suggestion. The phrase, “this study finding” (line 575) has been rephrased as “the 9.6% prevalence” (page 37, line 667). The phrase, “But this study's result” (611–612) has been rephrased as “However, this 13.7% prevalence” (page 39, line 711).

• “the present study's findings (page 32, line 569-570), has been rephrased as “this 9.6% prevalence” (page 37, line 662).

• This result (page 33, line 596), has been rephrased as “The 79.7% result” (page 38, line 695).

• This finding (page 34, line 600), rephrased as “This 79.7%” (page 38, line 699-700).

• This result (page 34, line 608), rephrased as “The prevalence of 13.7% MDD” (page 39, line 707).

Comment: • There are a few grammatical inconsistencies. A thorough proofreading of the manuscript is recommended to ensure clarity and coherence.

Response: Thank you for your feedback. Grammatical inconsistencies have been corrected throughout the manuscript to ensure clarity and coherence.

Comment: •Recommended course of action: Request Revision

Response: Thank you for your recommendation. We have revised the manuscript accordingly.

Reviewer 3

Major Comments

Comment: 1. Justification for Age Range (6–8 months)

While the focus on infants aged 6–8 months aligns with WHO recommendations for the introduction of complementary foods, the authors should clarify why they did not extend the age group to 23 months. This limits comparability with other IYCF studies that use the full 6–23 months range. Further justification would strengthen the rationale.

Response: Thank you for your insightful feedback. Further justification has been added for why this study particularly focused on mothers with infants aged 6–8 months, rather than extending to 23 months (please refer to pages 4, lines 67-68 and page 6, lines 123-132).

Comment: 2. Definition of ACFP and Indicator Selection

The use of revised 2021 WHO IYCF indicators, including breast milk as a food group, is commendable. However, the authors should explain the implications of this change on the comparability of their findings with studies using the 2008 guidelines.

Response: Thank you for highlighting this important aspect. The use of the revised WHO indicators (2021), other than the 2008 guidelines, emphasizes the inclusion of breastfeeding as one of the food groups in the MDD calculation. This change can affect MDD, MAD, and ACFP. In the current study, the prevalence of MDD was lower than study findings using the 2008 WHO IYCF practices indicators. This difference arises from the inclusion of breast milk as part of the eight food groups, compared to the calculation based on seven food groups. Evidence indicates that the MDD calculated based on the eight food group indicators is lower than the MDD-calculated using the seven food group indicators outlined in the 2008 WHO guidelines. Additionally, this study applied the new WHO guidelines to assess MDD, MAD, and ACFP, however, we referenced previous literature using the 2008 WHO guidelines in the discussion due to limited evidence supporting the new guidelines (we addressed this issue in the limitation section) (please refer to Pages 5, lines 89-94, page 39, lines 715-719 and page 44, lines 826-828).

Comment: • Additionally, the inclusion of egg/flesh food consumption as a separate indicator for ACFP needs further justification and clarity.

Response: Thank you for your feedback. Egg and flesh food consumption is one of the new complementary feeding indicators included in the revised WHO document for assessing IYCF practices. Eggs and meat are nutrient-dense food groups that are crucial for maximizing child nutrition and are vital for optimal linear growth (kindly refer to page 5, line 94-99).

Comment: 3. Low Prevalence of ACFP

The reported prevalence of only 9.6% is alarmingly low. The discussion would benefit from a deeper exploration of its implications for public health, particularly in the context of national programs such as the Seqota Declaration and SDG targets.

Response: Thank you for your important observation. We have addressed the implications of the current very lower prevalence of ACFP in the context of practical and policy implications including the Ethiopia’s national program, the “Seqota Declaration of Zero Hunger”, which aims to “end stunting in children under two”, as well as the global commitment particularly the SDG target 2.2, which seeks to “end all forms of malnutrition by 2030” (please refer to pages 37 & 38, lines 675-681).

Comment: 4. Accounting for Cluster Design

While a generalized linear mixed model is mentioned, the low intra-cluster correlation coefficient raises questions about the necessity of clustering in analysis. Please clarify how clustering was addressed statistically and whether simpler models were considered.

Response: Thank you for your insightful comment. Given the categorical nature of the outcome variable, a generalized linear mixed model (GLMM) was initially fitted to address cluster-level variables. The low intra-cluster correlation coefficient (ICC) indicates that most of the variance is within clusters rather than between them. However, we did not consider simpler models, such as generalized linear models (GLMs); instead, we used logistic regression (kindly refer to page 14, lines 292-298).

Comment: Also, explain the rationale behind selecting a design effect of 2.

Response: We multiplied the calculated sample size by a design effect of 2 to better account for the potential impact of clustering on estimates and to increase the overall sample size (please refer to page 14, lines 298-300).

Comment: 5. Measurement and Validation of Psychosocial Predictors

The manuscript reports “perceived susceptibility” as a significant predictor of ACFP. More detail is needed on how this and other psychological constructs (severity, barriers, benefits) were measured and whether the instruments were validated in this population.

Response: Thank you for your valuable comment. The predisposing factors were assessed using a validated tool adapted, in accordance to child feeding practices, from a previous study conducted in the rural setting in Dessie, Amhara regional state, North central Ethiopia (kindly refer to pages 12 & 13, lines 2645-275).

Comment: 6. Integration of Mixed Methods

Although both quantitative and qualitative components are well described, the manuscript would benefit from more explicit integration. A discussion section comparing, and contrasting quantitative findings with qualitative themes would improve cohesion.

Response: Thank you for your constructive feedback. We have tried to explicitly integrate the quantitative and qualitative findings in the discussion section to improve cohesion (Kindley refer pages 37, lines 657- 659; page 38, lines 683-685; page 39, lines 705-706; page 40, lines 742-747; page 41, lines 755-758 and 768-771; page 42, lines 782-788; page 42 & 43, lines 794-796; and page 43, lines 798-800).

Comment: 7. Wealth Index Construction

The authors used principal component analysis (PCA) to construct a wealth index. Please provide additional detail: • Variables included in the PCA

Response: Thank you for your insightful comment. After performing rotations to make the components more interpretable, eleven variables with a commonality value greater than 0.5 were retained to produce factor scores. These variables include, households with 1) radio, 2) dining table, 3) sofa or chairs, 4) bed with cotton or sponge mattress, 5) cattle (cows or bulls), 6) horses, donkeys, or mules, 7) chickens, 8) watch (by at least one household member), 9) animal-drawn cart (by at least one household member), 10) location of drinking water, and 11) proximity of drinking water from home (please refer to page 12, lines 246-253).

Comment: • Percent variance explained by the fir

---

## [Decision Letter · Decision Letter 1]

6 Jun 2025

Dear Dr. Birhanu,

Thank you for submitting your manuscript to PLOS ONE. After careful consideration, we feel that it has merit but does not fully meet PLOS ONE’s publication criteria as it currently stands. Therefore, we invite you to submit a revised version of the manuscript that addresses the points raised during the review process.

**Please submit your revised manuscript by Jul 21 2025 11:59PM. If you will need more time than this to complete your revisions, please reply to this message or contact the journal office at plosone@plos.org . **

**A rebuttal letter that responds to each point raised by the academic editor and reviewer(s). You should upload this letter as a separate file labeled 'Response to Reviewers'.****A marked-up copy of your manuscript that highlights changes made to the original version. You should upload this as a separate file labeled 'Revised Manuscript with Track Changes'.****An unmarked version of your revised paper without tracked changes. You should upload this as a separate file labeled 'Manuscript'.**

****

**If applicable, we recommend that you deposit your laboratory protocols in protocols.io to enhance the reproducibility of your results. Protocols.io assigns your protocol its own identifier (DOI) so that it can be cited independently in the future. For instructions see: https://journals.plos.org/plosone/s/submission-guidelines#loc-laboratory-protocols . Additionally, PLOS ONE offers an option for publishing peer-reviewed Lab Protocol articles, which describe protocols hosted on protocols.io. Read more information on sharing protocols at https://plos.org/protocols?utm_medium=editorial-email&utm_source=authorletters&utm_campaign=protocols .**

We look forward to receiving your revised manuscript.

**Kind regards,**

**Omnia Samir El Seifi, M.D., Ph.D.**

**Academic Editor**

PLOS ONE

**Journal Requirements:**

Reviewers' comments:

**Reviewer's Responses to Questions**

**Comments to the Author**

**Reviewer #2: All comments have been addressed**

**Reviewer #3: All comments have been addressed**

**2. Is the manuscript technically sound, and do the data support the conclusions??>**

**Reviewer #2: Yes**

**Reviewer #3: Yes**

**3. Has the statistical analysis been performed appropriately and rigorously? ?>**

**Reviewer #2: Yes**

**Reviewer #3: Yes**

**4. Have the authors made all data underlying the findings in their manuscript fully available??>**

**Reviewer #2: Yes**

**Reviewer #3: Yes**

**5. Is the manuscript presented in an intelligible fashion and written in standard English??>**

**Reviewer #2: Yes**

**Reviewer #3: Yes**

**Reviewer #2: (No Response)**

**Reviewer #3: Ensure figure legends and labels (especially for Figs 2 & 3) are clearly readable and interpretable without relying on the main text.**

**Consider adding a brief, summarizing paragraph at the end of the discussion explicitly comparing quantitative and qualitative findings.**

**what does this mean? ). If published, this will include your full peer review and any attached files.**

**Do you want your identity to be public for this peer review?** For information about this choice, including consent withdrawal, please see our Privacy Policy

**Reviewer #2: No**

**Reviewer #3: Yes: Muhammad Aasim**

****

**While revising your submission, please upload your figure files to the Preflight Analysis and Conversion Engine (PACE) digital diagnostic tool, https://pacev2.apexcovantage.com/ . PACE helps ensure that figures meet PLOS requirements. To use PACE, you must first register as a user. Registration is free. Then, login and navigate to the UPLOAD tab, where you will find detailed instructions on how to use the tool. If you encounter any issues or have any questions when using PACE, please email PLOS at figures@plos.org . Please note that Supporting Information files do not need this step.**

---

## [Author Response · Author response to Decision Letter 2]

15 Aug 2025

Response to reviewers

We sincerely appreciate your time and valuable feedback on our manuscript. This is our second revision, and we have thoughtfully addressed each of your comments in detail. Please note that the page and line numbers referenced in our responses correspond to those in the clean version submitted as “Manuscript.”

Editor

Journal Requirements:

Comment: Please review your reference list to ensure that it is complete and correct. If you have cited papers that have been retracted, please include the rationale for doing so in the manuscript text, or remove these references and replace them with relevant current references. Any changes to the reference list should be mentioned in the rebuttal letter that accompanies your revised manuscript. If you need to cite a retracted article, indicate the article’s retracted status in the References list and also include a citation and full reference for the retraction notice.

Response: Thank you for your comment regarding the reference list. We have carefully reviewed and revised the references list to ensure completeness and accuracy.

• In this revision, we removed duplicate references.

• We also added Digital Object Identifiers (DOIs) to make the references more complete.

• We have not cited any retracted articles, and the current version of the reference list contains no retracted publications. Thank you for highlighting this important aspect of manuscript quality.

Reviewer 3

Comment: Ensure figure legends and labels (especially for Figs 2 & 3) are clearly readable and interpretable without relying on the main text.

Response: Thank you for your feedback. We have carefully revised Figures 2 and 3 to ensure all percentages are accurate, legends, and labels are fully interpretable without reference to the main text. The specific improvements include:

Figure 2 revisions:

1. Added a clear, descriptive title,

2. Included “consumption categories” as a heading for Yes/No legends,

3. To ensure unambiguous interpretation, we explicitly defined 'Yes' as 'consumed' and 'No' as 'not consumed (kindly refer to Fig 2. (uploaded in our submission) and the figure caption on page 23, line 436).

Figure 3 revisions:

1. Incorporated a clear, descriptive title,

2. Included “Categories and sample sizes by age” as a heading for age classifications (6m, 7m, and 8m), along with the corresponding population size for each age group,

3. Calculation errors were corrected for child feeding indicators, including Minimum meal frequency, Minimum acceptable diet, Egg and/or flesh food consumption, Appropriate complementary feeding practices,

4. Replaced all abbreviations of indicators with full terms to enhance clarity (kindly refer Fig 3. (uploaded in our submission) and the figure caption on page 26, lines 469-470).

These improvements may ensure both figures now contain all necessary information for standalone interpretation.

Comment: Consider adding a brief, summarizing paragraph at the end of the discussion explicitly comparing quantitative and qualitative findings.

Response: Thank you for your valuable suggestion. We have added a brief summarizing paragraph at the end of the discussion section comparing the quantitative and qualitative findings (please refer to pages 41 and 42, lines 764-773).

Additional revision

We have carefully reviewed and revised the manuscript throughout to address:

• Grammatical and spelling errors, unnecessary spacing, and punctuation issues,

• Consistency in the use of abbreviations,

• Redundant or unclear sentences, to improve clarity, consistency/ flow, and readability.

In the qualitative section:

• Some quotes were conceptually shortened to enhance clarity and coherence while preserving their original meaning.

• To minimize the risk of identifying participants, quote attributions (speaker tags) were rephrased or removed.

---

## [Decision Letter · Decision Letter 2]

1 Oct 2025

Appropriate complementary feeding practices and associated factors among mothers with infants aged 6–8 months in West Gojjam Zone, Northwest Ethiopia: A mixed methods study

PONE-D-24-46708R2

Dear Dr. Birhanu,

We’re pleased to inform you that your manuscript has been judged scientifically suitable for publication and will be formally accepted for publication once it meets all outstanding technical requirements.

Kind regards,

Dinaol Abdissa Fufa, Mph

Academic Editor

PLOS ONE

Reviewers' comments:

Reviewer's Responses to Questions

**Comments to the Author**

Reviewer #2: All comments have been addressed

Reviewer #3: All comments have been addressed

2. Is the manuscript technically sound, and do the data support the conclusions?

Reviewer #2: Yes

Reviewer #3: Yes

3. Has the statistical analysis been performed appropriately and rigorously?

Reviewer #2: Yes

Reviewer #3: Yes

4. Have the authors made all data underlying the findings in their manuscript fully available?

Reviewer #2: Yes

Reviewer #3: Yes

5. Is the manuscript presented in an intelligible fashion and written in standard English?

Reviewer #2: Yes

Reviewer #3: Yes

Reviewer #2: (No Response)

Reviewer #3: The authors have adequately addressed the concerns raised in my previous review:

Figures 2 and 3: The figures have been revised with clear, descriptive titles, explicit labeling of categories, corrected calculations, and replacement of abbreviations with full terms. These improvements make the figures self-explanatory and interpretable without reliance on the main text.

Discussion Section: A summarizing paragraph has been added at the end of the discussion (pp. 41–42, lines 764–773), explicitly comparing the quantitative and qualitative findings. This addition enhances integration of the study’s mixed-methods approach.

Additional Improvements: The authors also undertook general editorial revisions, including corrections of grammar and abbreviations, removal of redundancies, and ethical adjustments to qualitative quotes. These strengthen clarity, readability, and protection of participant confidentiality.

Overall Assessment:

The authors have satisfactorily complied with my comments. The revisions improve both the clarity of presentation and the interpretability of the findings.

**Do you want your identity to be public for this peer review?** For information about this choice, including consent withdrawal, please see our Privacy Policy

Reviewer #2: No

Reviewer #3: **Yes: ** Dr. Muhammad Aasim

---

## [Editor Report · Acceptance letter]

PONE-D-24-46708R2

PLOS ONE

Dear Dr. Birhanu,

I'm pleased to inform you that your manuscript has been deemed suitable for publication in PLOS ONE. Congratulations! Your manuscript is now being handed over to our production team.

Kind regards,

on behalf of

Dr. Dinaol Abdissa Fufa

Academic Editor

PLOS ONE